# Persistent Maternal Mental Health Disorders and Toddler Neurodevelopment at 18 Months: Longitudinal Follow-up of a Low-Income South African Cohort

**DOI:** 10.3390/ijerph20126192

**Published:** 2023-06-20

**Authors:** Marlette Burger, Christa Einspieler, Esme R. Jordaan, Marianne Unger, Dana J. H. Niehaus

**Affiliations:** 1Faculty of Medicine and Health Sciences, Physiotherapy Division, Department of Health and Rehabilitation Sciences, Stellenbosch University, Cape Town 8000, South Africa; 2Research Unit iDN—Interdisciplinary Developmental Neuroscience, Division of Phoniatrics, Medical University of Graz, 8010 Graz, Austria; 3Biostatistics Unit, South African Medical Research Council, Cape Town 7500, South Africa; 4Statistics and Population Studies, University of the Western Cape, Cape Town 8000, South Africa; 5Faculty of Medicine and Health Sciences, Department of Psychiatry, Stellenbosch University, Cape Town 8000, South Africa

**Keywords:** persistent maternal mental health disorders, maternal psychosis, maternal mood disorders, maternal comorbid anxiety and mood disorders, toddler neurodevelopment, South Africa

## Abstract

One of the biggest threats to early childhood development in Africa is poor maternal mental health. The present study reports on the relationships between clinical diagnoses of persistent maternal mental health disorders (at 3- and/or 6- and 18-month post-term age) and toddler neurodevelopment at 18 months of age. Eighty-three mother–toddler dyads from low socio-economic status settings in Cape Town, South Africa, were included. At the 3-, 6- and 18-month postnatal visits, clinician-administered structured diagnostic assessments were carried out according to the Diagnostic and Statistical Manual of Mental Disorders-V (DSM-V) criteria. Toddler neurodevelopment at 18 months corrected age was assessed with the Bayley Scales of Infant and Toddler Development (BSID-III). No significant differences (*p* > 0.05) were found between toddlers with exposure to persistent mood or psychotic disorders in the different BSID-III domains compared to toddlers with no exposure. Toddlers exposed to persistent comorbid anxiety and mood disorders scored significantly higher on the cognitive (*p* = 0.049), motor (*p* = 0.013) and language (*p* = 0.041) domains and attained significantly higher fine motor (*p* = 0.043) and gross motor (*p* = 0.041) scaled scores compared to toddlers with no maternal mental health disorder exposure. Future investigations should focus on the role of protective factors to explain the pathways through which maternal mental health status is associated with positive toddler neurodevelopmental outcomes.

## 1. Introduction

The transition from early infancy to toddlerhood is a time of remarkable brain development and growth. A child’s neurodevelopment encompasses several developmental domains, namely cognitive, language, motor, social-emotional and adaptive behaviour [1]. The success with which early neurodevelopmental milestones are met depends on a combination of the child’s genetic predisposition, his/her capacity to tolerate the environment and the quality of the early caregiving relationship [2]. Given the extreme demands involved in raising healthy children in Africa [3,4,5], a nurturing home environment with a sensitive and responsive caregiver may be a vital moderator in lessening the undesirable effects of socio-economic adversity on the physical development and neurodevelopment of the child [6,7]. In South Africa, where 60% of children aged 0 to 4 years are multidimensionally poor and deprived of adequate housing, health care, early childhood education and WASH (water, sanitation and hygiene), a nurturing home environment is particularly important [8]. 

One of the biggest threats to a nurturing home environment and early childhood development is poor maternal mental health [6,9]. According to the Diagnostic and Statistical Manual of Mental Disorders-V (DSM-V) [10] criteria, mental health disorders are defined as “a syndrome characterized by clinically significant disturbance in an individual’s cognition, emotion regulation, or behavior that reflects a dysfunction in the psychological, biological, or developmental processes underlying mental functioning”. Throughout the early years, the mother’s well-being and the health and development of her child are closely intertwined. Maternal mental health disorders can seriously disrupt maternal nurturing caregiving behaviours and reduce maternal sensitivity and responsiveness [11], which may have a profound effect on the extent of nurturing in the child’s home environment and her/his neurodevelopment [2,12]. Compelling scientific evidence derived from high-income countries (HICs) confirms that the persistence of mental health symptoms of mothers throughout infancy has been associated with unfavorable outcomes in various domains of development in young children [13,14,15,16]. An important factor that may increase the risk of unfavorable child developmental outcomes is the persistence and severity of mental health disorders during the postnatal period, especially for mother–child dyads exposed to socio-economic adversities [17]. However, relatively little work has been carried out in low- and middle-income countries (LMICs) where psychosocial and environmental risk factors differ substantially from those in HICs [18,19]. Most studies carried out in LMICs on the effect of maternal mental health disorders on infant and toddler development are cross-sectional or, in the case of longitudinal studies, only include one prior assessment of maternal mental health disorders [20], therefore not taking the persistence of maternal mental health disorders over time into account.

The current study aimed to address this gap by assessing the relationships between distinct clinical diagnoses of persistent maternal mental health disorders (at 3- and/or 6- and 18-month post-term age) and toddler neurodevelopment across five developmental domains at 18 months of age. This is a longitudinal follow-up study of a cohort of mother–infant dyads living in poor urban and peri-urban communities in Cape Town, South Africa. Our previous analyses revealed no significant differences on any of the domains of the Bayley Scale, Third Edition (BSID-III) [21] domains at 6 months post-term between infants exposed to persistent maternal mood disorders only and infants exposed to persistent comorbid disorders (i.e., a combination of mood, anxiety and psychotic disorders) at 3 and 6 months compared to unexposed infants [22]. However, infants exposed to persistent psychotic disorders at 3 and 6 months scored significantly lower on the cognitive and motor domains of the BSID-III [22].

The objectives of the current study were to examine the associations between persistent (i) maternal mood disorders; (ii) maternal psychotic disorders; and (iii) comorbid maternal anxiety and mood disorders (at 3 and/or 6 months and 18 months) and the motor, cognitive, language, social-emotional and adaptive behavioural domains of 18-month-old toddlers.

## 2. Materials and Methods

### 2.1. Design and Setting

The current longitudinal study formed part of a larger prospective cohort design, the Maternal and Infant Mental Health (MIMH) study. The MIMH study recruited participants from Stikland Psychiatric Hospital, a specialist psychiatric facility that provides services to largely low socio-economic communities, and a private-funded Well Baby follow-up clinic in the northern suburbs of Cape Town, South Africa. The data for the current sub-study are derived from Stikland Hospital. Neurodevelopmental follow-up assessments of the 18-month-old toddlers included in the current study were conducted at 3- and 6-months corrected age, and the procedures and findings have been reported elsewhere [22,23]. The procedures applicable to the follow-up assessments of mother–child dyads at Stikland Hospital are described below.

### 2.2. Participants

Pregnant women aged 18 years and older attending the Maternal Mental Health Outpatient Clinic at Stikland Hospital with a previous or current diagnosis of psychiatric illness [according to the DSM-V [10] criteria] were eligible for the current study. South Africa has a linguistically and culturally heterogeneous demographic profile, and women attending this health facility are mainly Afrikaans, English or isiXhosa speaking. Mothers who provided informed consent were enrolled in their first, second or third trimester of pregnancy. Toddlers born between 1 April 2014 and 30 September 2019 to mothers with a history of psychiatric illnesses and a confirmed DSM-V diagnosis at 3 and/or 6 months and 18 months were eligible for inclusion in the study. Toddlers that had prenatal or postnatal (through breastfeeding) exposure to maternal psychotropic medication and/or medication for other conditions, such as diabetes mellitus, epilepsy or human immunodeficiency virus (HIV) infection, were also eligible for inclusion. Toddlers born preterm and/or having low birth weight and therefore presented with a risk for developmental delays were not excluded from the study. The reason for including these at-risk toddlers is that maternal mental health disorders are associated with unhealthy maternal lifestyles and adverse pregnancy, birth and infant health outcomes in Africa, such as preterm birth and/or low birth weight [24]. However, toddlers with the following disorders or conditions (and therefore a higher risk for developmental delays) were excluded: (i) deformities of the central nervous system (e.g., myelomeningocele); (ii) congenital disorders (e.g., arthrogryposis multiplex congenital); (iii) microcephaly (<3rd percentile for gestational age and sex); (iv) diagnosis of genetic disorders (e.g., Down syndrome); (v) hypoxic-ischemic encephalopathy; (vi) traumatic brain injury; (vii) foetal alcohol syndrome; (viii) significant hearing or vision impairment; (vix) any serious medical condition (e.g., heart failure, tuberculous meningitis and cystic fibrosis); and (x) out-of-home placement/foster care.

In the comparison group, the same inclusion and exclusion criteria applied, and only mother–child dyads with no antenatal or postnatal exposure to mental health illnesses were included. The comparison group resided in the same low-income communities as the rest of the study cohort. Obstetric nurse practitioners at community health clinics and day hospitals recruited the comparison group within the first week after the infants were born. Since no previous maternal medical history records were available, the prenatal medical and psychiatric history was obtained from the mothers. Trained research assistants obtained written informed consent from mothers in their language of choice. Informed consent was obtained during the enrolment of mothers at Stikland Hospital.

### 2.3. Measures

#### 2.3.1. Socio-Demographic and Health Characteristics of the Mothers and Toddlers

Trained research staff collected comprehensive health data from maternal and toddler hospital records and the Children’s Road to Health booklets [25] and from interviews with the mothers. Socio-demographic variables were collated antenatally and reviewed at the 3-, 6- and 18-month postnatal visits using a structured interviewer-administered questionnaire in the participants’ preferred language. The socio-demographic variables of interest at the 18-month postnatal assessment included maternal age, educational achievement (completed primary, secondary or tertiary education), antenatal physical illness (HIV status), antenatal and postnatal substance use, psychotropic medication use, marital status, employment status (employed or unemployed), average household income (<300 USD pm; >300 USD pm), infant birth and toddler (18-month) health outcomes, caregiver during the day and number of siblings. During the 18-month assessment, the toddler’s weight, maternal reports of the toddler’s health, hospital records of the toddler’s health and reasons for hospitalisations since birth were recorded. 

#### 2.3.2. Assessment of Child Neurodevelopment: Bayley Scales of Infant and Toddler Development

The BSID-III was used to assess neurodevelopment at 18 months post-term age [21]. The BSID-III is currently considered the gold standard worldwide [26], and The World Bank recommends the BSID be used as the primary tool for assessing child development in LMICs [27]. The BSID-III is validated for use in healthy Black African urban infants aged 2–13 months in South Africa [28]. Another study reported that healthy toddlers residing in urban areas in South Africa, with a mean age of 19.4 months, scored slightly below the 50th percentile for the BSID-III USA reference population but still within the normal range on the different developmental domains of the BSID-III [29]. 

A trained paediatric physiotherapist (the first author) administered the BSID-III. She was blinded to the maternal mental health diagnoses and the toddlers’ prenatal, perinatal and postnatal medical history. The cognitive, motor and language domains were assessed by direct observation in the child’s preferred language (Afrikaans, English or isiXhosa) with instructions consistent with administration and scoring directions set out by the manual. The social-emotional and adaptive behaviour subscales were assessed through observation of the toddler and maternal reports. Assessments were conducted in a standard testing environment (an uncluttered, comfortable room in a quiet setting) with ample space for the toddlers to demonstrate gross motor skills such as running, stair climbing and jumping. Only the assessor, the toddler and the mother were present during the assessment to minimise any distractions. It took 60 to 90 min to administer the five domains of the BSID-III. We scheduled assessments to fit into the child’s sleeping and eating patterns to avoid factors such as tiredness or hunger influencing test results. Testing was rescheduled within the same week if it was impossible to administer the tests within one session because the child was fatigued, restless and/or upset. The BSID-III provides composite scores for cognitive, motor and language skills as well as social-emotional and adaptive behaviour (range: 40–160; mean = 100, SD = 15). Separate scaled scores are provided for fine and gross motor and expressive and receptive language subscales (range: 1–19; mean = 10, SD = 3). Composite scores were categorised as above-average (standardised score > 115), average (>85), delayed (70–84) or severely impaired (<70) performance. Scaled scores were classified as above-average (>13), average (8–12) and below-average (≤7) performance. Toddlers scoring ≤ 1 SD below the mean of the composite (<85) or scaled (≤7) scores were classified as demonstrating a clinically significant developmental delay. Except where otherwise stated, motor development refers to combined gross and fine motor development and language development to combined expressive and receptive language development.

#### 2.3.3. Assessment of Maternal Mental and Physical Health

At the 3-, 6- and 18-month postnatal visits, clinician-administered structured diagnostic assessments as per standard of care were carried out to assess maternal mental health. A qualified senior psychiatrist with at least five years’ experience conducted the assessments in English or Afrikaans. In cases where IsiXhosa-speaking women were unable to communicate in either English or Afrikaans, a trained translator assisted. Psychiatric diagnoses were made according to the DSM-V diagnostic criteria [10]. Additional psychiatric screening tools were also used in the assessments, namely, the Edinburgh Postnatal Depression Scale (EPDS) [30], the Mini International Neuropsychiatric Interview (MINI) [31] and the Recent Life Events Questionnaire (RLEQ) [32]. Mothers were grouped according to their psychiatric diagnosis as having either (i) mood disorders, (ii) psychotic disorders, (iii) comorbid anxiety and mood disorders or (iv) no psychiatric disorder (comparison group). Current physical health and general medical history were assessed and documented during the semi-structured interviews. Smoking (tobacco), alcohol and recreational drugs were also recorded during the ante- and postnatal period with a dichotomous measure (exposure versus no exposure). The frequency of substance use was not recorded.

### 2.4. Description of Mother–Toddler Participants

Figure 1 presents the distinct groups of mother–toddler dyads defined by clinical diagnoses of maternal mental health disorders across 3 time points from early post-term to 18 months post-term. Of the 97 dyads, 14 were excluded because the mothers had a psychiatric diagnosis at only one time point (e.g., at 3 or 6 months, n = 5; or at 18 months, n = 1) or mother–toddler dyads were lost to follow-up (n = 8). Eighty-three mother–toddler dyads were included in the final sample: 31 toddlers with no exposure to maternal psychiatric disorders (comparison group), 30 toddlers with exposure to maternal mood disorders at all three time points, 13 toddlers with exposure to psychotic disorders (at two of the three time points) and 9 toddlers with exposure to anxiety and/or mood disorders (at two of the three time points). To be included in the mood or psychotic diagnostic groups, mothers had to be diagnosed with a mood or psychotic disorder at two of the three time points. The only mother with a comorbid diagnosis of psychotic, mood and anxiety disorders was included in the psychotic group. Since only 2 mothers presented with an anxiety disorder only at all three time points, they were included in the comorbid anxiety and mood disorder group. All the mothers with a postnatal psychiatric diagnosis (n = 52) had a history of previous mental health disorders (e.g., during and/or before their pregnancy), while none of the mothers in the comparison group reported any previous psychological symptoms. Mood disorders included bipolar disorder I and II and major depressive disorder, while psychotic disorders included schizophrenia and psychosis. Anxiety disorders included generalized anxiety disorder, panic disorder, agoraphobia, social phobia and posttraumatic stress disorders.

### 2.5. Ethical Considerations

Approval for this study was obtained by Stellenbosch University’s Health Research Ethics Committee (S12/04/111). Permission was obtained from the study’s site (Stikland Psychiatric Hospital) and the Western Cape Provincial Health Research Committee (Cape Town, South Africa). All mothers provided written informed consent and were informed that they could exit the study without affecting the further standard of care.

### 2.6. Statistical Analysis

Statistical analysis was performed using the Statistical Analysis System (SAS) 9.4 software, SAS Institute Inc., Cary, NC, USA. Maternal and child socio-demographic and health characteristics were described for the mental health disorder groups using either medians, lower and upper quartiles or numbers (%), whichever were appropriate. Similarly, descriptive statistics were provided for the BSID-III scales, which comprised composite scores for the motor, cognitive, language, social-emotional and adaptive behavioural domains and scaled scores for the fine and gross motor and the expressive and receptive language subscales. Due to small samples for two of the maternal mental health disorder groups (n = 13 for psychotic disorder and n = 9 for comorbid anxiety and mood disorders), Kruskal–Wallis tests were conducted to compare the BSID-III composite and scaled scores between the maternal mental health disorder groups and the comparison group. Furthermore, due to small sample sizes, it was not possible to conduct modelling with relevant mother–toddler socio-demographic and health covariates as confounders.

## 3. Results

### 3.1. Socio-Demographic and Health Characteristics of the Mothers and Toddlers

The socio-demographic and health characteristics of the mothers and toddlers who participated are presented in Table 1 and Table 2. All the mothers included in the cohort had completed their secondary education. More than 50% of the mothers in the comparison group and maternal mental health disorder groups were unemployed at 18 months post-term. The monthly family income was below the equivalent of 300 USD for most of the mother–toddler dyads. The mothers who earned an income were either employed in the formal sector or were self-employed (e.g., selling goods from their homes or working as street vendors). Most of the self-employed mothers were able to care for their toddlers during the day. Most of the mothers diagnosed with mood (73.3%) and psychotic (92.3%) disorders were using various classes of psychotropic medication at 18 months post-term, while 44% of mothers diagnosed with comorbid anxiety and mood disorders were using psychotropic medication. 

None of the toddlers exposed to HIV during pregnancy were HIV infected. None of the toddlers in the comparison group were born preterm (<37 weeks gestation), while four toddlers exposed to mood (13.3%), one toddler exposed to psychotic (7.7%) and one toddler exposed to comorbid anxiety and mood (11.1%) disorders were born moderately to late preterm (gestational age range: 34–36 weeks). Three toddlers had a history of postnatal hospitalisation before three months post-term age; one with an Escherichia coli infection (exposed to psychotic disorders), one with a respiratory tract infection (exposed to mood disorders) and one with jaundice (exposed to anxiety disorders). Apart from one toddler (exposed to psychotic disorders) that presented with chronic tonsilitis, the medical history of the toddlers was unremarkable, and none had been hospitalised between 3 and 18 months post-term age. None of the toddlers’ weight-for-age z score (WAZ) or head circumference-for-age z score (HCAZ) was below −2 at 18 months post-term age. 

### 3.2. Toddler Neurodevelopmental Outcome at 18 Months Post-Term Age

Table 3 portrays the results of the BSID-III composite and scaled scores for the following subgroups: toddlers with no exposure to maternal mental health disorders (comparison group; n = 31), toddlers with exposure to mood disorders (n = 30), toddlers with exposure to psychotic disorders (n = 13) and toddlers with exposure to comorbid anxiety and mood disorders (n = 9). At 18 months, toddlers exposed to persistent comorbid anxiety and mood disorders attained significantly higher cognitive (*p* = 0.049), motor (*p* = 0.013) and language (*p* = 0.041) composite scores as well as fine motor (*p* = 0.043) and gross motor (*p =* 0.041) scaled scores compared to toddlers with no maternal mental health disorder exposure. No significant differences (*p* > 0.05) were found between toddlers with exposure to persistent maternal mood or psychotic disorders on the different BSID-III domains and subscales and toddlers without exposure. None of the included toddlers scored below 85 (delayed performance) or 70 (severely impaired) on the social-emotional composite score.

The number of toddlers (n = 15) that presented with a clinically delayed and/or severely impaired performance on the various domains and subdomains of the BSID-III are summarised in Table 4. None of the toddlers exposed to comorbid anxiety and mood disorders demonstrated a clinically delayed and/or severely impaired performance on any BSID-III domains or subscales. Four toddlers in the comparison group (12.9%) presented with a below-average (≤7) score on the gross motor subscale, while two of these toddlers also presented with a delayed performance on the adaptive behaviour composite score. Eight toddlers exposed to maternal mood disorders (26.7%) demonstrated a clinically delayed and/or severely impaired performance on various BSID-III domains or subscales. One of the toddlers exposed to mood disorders demonstrated a clinically delayed and/or severely impaired performance on all the BSID-III domains and subscales except for the social-emotional domain. Three toddlers exposed to maternal psychotic disorders (23.1%) demonstrated a clinically delayed and/or severely impaired performance on various BSID-III domains or subscales. The only toddler exposed to three maternal psychiatric disorders (psychotic and mood and anxiety) at all three time points scored average or above average on all the BSID-III domains and subdomains.

## 4. Discussion

The current prospective study is one of the first to investigate whether clinical diagnoses of persistent mental health disorders of mothers residing in low-income communities in Cape Town, South Africa, were associated with various neurodevelopmental domains in 18-month-old toddlers. No significant differences were found between toddlers with and without exposure (comparison group) to persistent and concurrent maternal mood disorders (n = 30), as measured at 3 and/or 6 months and 18 months post-term on any of the domains and subscales of the BSID-III. This contrasts with earlier and recent findings in HICs. Cornish et al. (2005) reported significantly lower motor and cognitive performance on the BSID-II in 15-month-old toddlers exposed to persistent mood disorders throughout the first 12 months post-term and beyond [33]. The authors, however, did not find any adverse effects of persistent exposure to mood disorders on toddler language development [33]. A study conducted in Canada found that toddlers of mothers who had a trajectory of increasing depressive symptoms during the first postpartum year (measured at 6 weeks and 3, 6 and 10 months) showed more internalising and externalising behavioural problems at three years of age [34]. A study conducted in Brazil found that maternal depression at 1–2 months and 12 months postpartum was significantly associated with lower scores on the BSID-III language domain of toddlers at 12 months of age [35]. Interestingly, the expressive language subscale was also the subscale on which most of the toddlers in the current study with a clinical developmental delay presented with the worst scores (see Table 4). Hardly any studies have investigated the effect of exposure to maternal mood disorders beyond the first postnatal year in LMICs. However, our findings are consistent with a recent study conducted in low-income communities in Cape Town, adding to sparse evidence from South Africa. Garman et al. (2019) found that motor and cognitive outcomes on the BSID-III of 18-month-old toddlers and social-emotional or behavioural outcomes at 36 months postpartum did not differ across persistent high and low maternal depressive symptom trajectories as measured at 2 weeks and 6 and 18 months postpartum [36]. 

Although toddlers exposed to maternal psychotic disorders (n = 13) presented with the lowest scores on the cognitive, motor and language domains, they did not score significantly lower on any of the domains and subscales of the BSID-III compared to toddlers with no exposure. Previously, we reported that 6-month-old infants with exposure at 3 and 6 months to persistent psychotic disorders scored significantly lower on the motor and cognitive domains of the BSID-III [22]. Our current and previous findings are in accordance with previous studies conducted in low-income, one-parent households in the USA [37] and, more recently, in China [38]. These authors demonstrated short-term and transient delays in the cognitive [37,38], motor, social-emotional and adaptive behavioural domains [38] of infants exposed to maternal schizophrenia. Most of the mothers with psychosis in the current study were mentally ill during pregnancy and were treated with atypical antipsychotic medication during their pregnancy (data not provided). According to Peng et al. (2013), atypical antipsychotic treatment during pregnancy may be protective on the neurodevelopment of the children of mothers with psychosis, but the authors cautioned that case–control studies and longer follow-up beyond the first few postnatal years were warranted [38].

Contrary to expectations, the small group of toddlers exposed to persistent anxiety disorders (n = 9) attained significantly higher cognitive, motor and language composite scores and fine motor and gross motor scaled scores compared to toddlers with no maternal mental health disorder exposure. These findings were unexpected since seven of the nine mothers had received a diagnosis of comorbid anxiety and mood disorders, while two mothers had been diagnosed with anxiety disorders only. This is clinically important since experiencing comorbid anxiety and mood disorders such as depression involves more severe and persistent symptomatology and functional impairment than experiencing either disorder alone [39,40]. Furthermore, comorbidity is also associated with poorer mother–infant interactions, difficult infant temperament and negative consequences for the mother–infant relationship [41,42], which in turn may negatively affect toddler neurodevelopment [43]. 

Even though the literature suggests that comorbid anxiety and depression may have a negative impact on infant development [44], the potential link between the co-occurrence of persistent maternal anxiety and mood disorders and infant or toddler neurodevelopment is a new field of research and has been explored far less. Our findings differ from that of Ali et al. (2013), who reported that exposure to persistent maternal anxiety and depressive disorders based on DSM-IV criteria (assessed at 1, 2, 6, 12 and 18 months) had a significant adverse impact on the cognitive, language, gross and fine motor and emotional developmental domains of toddlers in Pakistan [45]. Contrary to the findings of Ali et al. (2013) [45], another large study from Pakistan found that persistent high maternal depressive and anxiety symptoms (assessed at 6, 12, 18 and 24 months postpartum) were not associated with cognitive, language and motor development on the BSID-III at 24 months of age after adjusting for several risk factors such as the caregiving environment, mother–child interactions, socio-economic status and food security [46]. These researchers did, however, report a significant negative association between persistent high maternal anxiety and depressive symptoms and toddler social-emotional development [46]. An Ethiopian population-based study reported that elevated levels of persistent maternal depressive, anxiety and somatic symptoms (assessed at 2 and 12 months postpartum) were not significantly associated with 12-month-old toddlers’ motor, cognitive and language outcomes on the BSID-III after adjusting for confounding variables [47]. In accordance with the present results, a study conducted in the USA reported positive associations between increased maternal stress and cognitive development and expressive language and between increased persistent depressive symptoms and fine motor skills of 12-month-old toddlers. The authors also reported that toddlers exposed to the lowest and highest levels of postpartum perceived stress and depressive symptoms demonstrated accelerated receptive language ability. Furthermore, persistently high levels of anxiety, depressive and stress symptoms were associated with accelerated gross motor ability. Only maternal anxiety was associated with slightly lower cognitive scores [48]. 

Methodological differences between the current study and earlier longitudinal studies on the effect of persistent maternal mental health disorders and toddler outcomes may explain the contrasts in findings. Previous studies have based their findings on standardised self-reported questionnaires or screening tools for depressive and/or anxiety symptoms [34,35,36,46,47,48,49] and/or maternal reports of toddler developmental outcomes [34,45]. The strengths of our study are that only women with a clinical psychiatric diagnosis were included and that a trained paediatric physiotherapist administered the cognitive, motor and language BSID-III domains. Furthermore, social-emotional and adaptive behaviour subscales were assessed by observing the toddlers and maternal reports. There is always the likelihood that mothers with poor mental health may not provide a reliable report of their toddler’s development (they may be positively biased, feel the need to give socially desirable answers or rate their toddler’s development poorer). Therefore, to strengthen study findings, maternal reports of domains such as social-emotional and adaptive behaviour should be based on multiple informants [50]. Previous studies using self-reporting screening tools to measure maternal symptoms of depression and anxiety were unable to conclude that toddler neurodevelopmental outcomes may be attributable to maternal anxiety, maternal depression or the comorbidity of the two disorders [45,46,47]. Since comorbidity rates of anxiety and mood disorders have been found to be very high [40], it is important that future studies should clearly define whether the effect on toddler neurodevelopmental outcome is due to maternal depression or anxiety alone or due to comorbidity of the two disorders.

There is a strong relationship between poor nutritional status and adverse child neurodevelopment. A meta-analysis reported a significant association between maternal depressive symptoms (clinically diagnosed or not) and early childhood underweight and stunting in LMICs [51]. Furthermore, in LMICs, children exposed to mothers with persistent depression are particularly at risk of stunting or being underweight [17]. Toddlers in the current study were not underweight (WAZ of −2 or below), which may explain why we did not find an association between persistent maternal mental health disorders and adverse neurodevelopment. Given that none of the toddlers were underweight, it may have served as a moderating factor since there is a strong relationship between poor nutritional status (stunting and underweight) and adverse cognitive and motor development [52].

All mothers with psychiatric diagnoses in our study attended Stikland Maternal Mental Health Outpatient Clinic and received a patient-specific therapeutic intervention. Although none of these interventions were specifically geared to improving mother–child relationships (e.g., maternal sensitivity and responsiveness and secure attachment) or parenting behaviour and practices, mothers may have developed coping strategies and support structures that could perhaps have a constructive impact on the development of their children in the presence of poor mental health. Some of these coping strategies or modifiable factors that may have a buffering effect against developmental delays in the presence of poor maternal mental health are dependable and healthy interpersonal relationships, reliable social support networks, maternal optimism, higher education and effective strategies to balance work, household and family responsibilities [53]. The fact that toddlers were monitored since early infancy, the high levels of clinical surveillance and the additional attention that the mothers and their toddlers received by the researchers might have acted as an additional form of support, buffering the effect of maternal mental health symptoms on toddler development [54]. Therefore, it could be argued that participating in the research study created in the mothers a heightened motivation and awareness of their responsibilities with regard to their children’s needs. Our study was not designed to analyse the mechanisms through which maternal mental health disorders were associated with toddler neurodevelopmental outcomes. It was, therefore, difficult to explain why toddlers with exposure to maternal comorbid anxiety and mood disorders scored significantly higher in the cognitive, motor and language domains. It may well be that this group of mothers had access to strong support networks and that they were able to provide a nurturing home environment for their toddlers’ development. Maternal depression and anxiety show unique features in symptomatology and are commonly expressed in two polarised mother–toddler interaction styles and maternal modelling behaviours. Mothers with depression exhibit passive, withdrawn and nonresponsive behaviour, while anxious mothers show intrusive, overstimulating, overprotective and controlling behaviour [55]. In comorbid anxiety and mood disorders, it is hard to differentiate between the combined effects of anxiety and mood disorders on mother–toddler interaction. It may be that toddlers exposed to comorbid anxiety and mood disorders in the current study were also exposed to overprotective behaviours. These overprotective maternal behaviours (hypervigilance) may have benefited the toddlers by compensating for or buffering the effect of maternal mood disorders, resulting in significantly higher motor, cognitive and language outcomes at 18 months. However, more research is needed to explore protective factors in the presence of persistent maternal mental health disorders. Identifying protective factors will inform strategies for supporting mothers with persistent mental health disorders and optimise children’s development.

Although typical for studies including a cohort with a clinical diagnosis of psychiatric disorders, a limitation of our study was the small sample size, in particular, the maternal comorbid anxiety and mood disorder and psychosis subgroups; therefore, our findings should be treated as preliminary. Due to the small sample sizes of the different subgroups, it was not possible to analyse socio-demographic and health characteristics as potential confounders in the pathway of toddler neurodevelopmental outcomes. There are higher rates of living in extended households, which include multigenerational family structures, in poorer areas in South Africa [56]. In fact, 62% of South African children live in extended households and kinship care for children in poorer areas in South Africa remains common [56]. In the current study, extended family members could have taken over the mothering role and provided a nurturing home environment for the infants and toddlers. Since 42% of fathers are absent from households in South Africa [57] and caring for the young infant lies primarily with the mother [58], the study focused only on maternal mental health. In the current study, 48% (n = 40) of the mother–toddler dyads were not living with the paternal father. However, the positive involvement of paternal fathers and men taking on father roles and responsibilities for non-biological children should not be underestimated and should be explored in future research. Therefore, one of the main limitations of the current study was that we did not assess social support structures at home or in the community. Furthermore, we included a clinical sample with a history of psychiatric disorders (diagnosed during and/or before their pregnancy). However, we were unable to control for or distinguish the effect of persistent mental health disorders prior to the postpartum period. Previous findings showed that prenatal stress was significantly related to higher BSID-III motor domain scores of 12-month-old toddlers after adjusting for several potential confounders, such as postnatal maternal and paternal depressive symptoms and antidepressant use [59]. Likewise, DiPietro et al. (2006) reported that increased levels of prenatal depressive and anxiety symptoms were associated with enhanced mental, motor and behavioural development in 2-year-old children after controlling for postnatal maternal anxiety, depression and stress [60]. Therefore, it is plausible that advanced toddler cognitive, motor and language development in the current study may be linked to prenatal anxiety, depression or stress through various biological pathways [60]. These biological pathways may include activation of the neuro-endocrine (hypothalamic-pituitary-adrenal) and sympathetic-adrenomedullary axes, with the subsequent placental transfer of glucocorticoids or neurotransmitters, which may result in changes in brain structure and function of the foetus [60]. We included mothers from low-income communities with a clinical diagnosis of maternal mental health disorders from a single medical centre. Therefore, our findings should not be generalised to other settings and populations until corroborated by larger longitudinal studies. Another limitation of our study was that we assessed neurodevelopmental outcomes in early childhood. Although the BSID-III was not designed to be a predictive tool, a handful of studies have assessed the ability of the BSID-III at 15–30 months post-term age to predict the neurodevelopmental outcome at 4–6.5 years of age [61,62,63,64]. Most of these studies reported that the BSID-III was not a good predictor for future cognitive function [63,64,65] or motor impairments other than cerebral palsy [61]. This highlights the need for longitudinal follow-up of similar cohorts to assess any lasting influence of exposure to persistent maternal mental health disorders.

## 5. Conclusions

The present study contributes to the limited research in LMICs on the neurodevelopmental outcomes of toddlers exposed to a clinical diagnosis of persistent maternal psychopathology in Africa. Our study presents novel evidence of the effect of persistent mood, psychotic and comorbid anxiety and mood disorders on multiple domains of toddler neurodevelopment. The findings that persistent exposure to mood, psychotic and comorbid anxiety and mood disorders did not have an adverse impact on toddler cognitive, motor and language development and toddler social-emotional and adaptive behaviour are reassuring for women with a clinical diagnosis of mental health disorders in our setting. However, due to the small sample sizes of maternal mental health disorder subgroups and the paucity of comparable research studies on this topic, it is much too soon to conclude that toddlers exposed to persistent mental health problems will not encounter neurodevelopmental problems in other settings and populations or at a later stage. Therefore, follow-up of similar larger cohorts is important to study the short- and long-term effects of exposure to persistent mental health disorders on the neurodevelopmental outcome of toddlers and older children. Since we found that toddlers exposed to comorbid anxiety and mood disorders presented with significantly higher cognitive, motor and language composite scores, future comprehensive investigations should focus on the role of possible protective or risk factors to understand, recognize and explain the pathways through which maternal mental health status is associated with toddler neurodevelopmental outcomes. Since extended households may play a significant role in the care of young infants and children in LMICs, future research should focus on the role of extended family members in providing a nurturing home environment for infants and toddlers.

## Figures and Tables

**Figure 1 ijerph-20-06192-f001:**
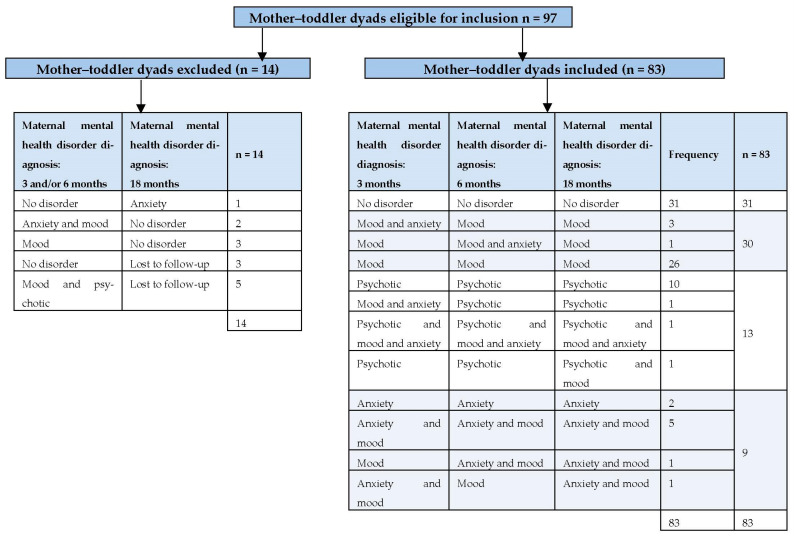
Flow diagram of mother–toddler dyads excluded and included according to the DSM-V diagnostic criteria.

**Table 1 ijerph-20-06192-t001:** Maternal socio-demographic and health characteristics (n = 83).

Maternal Demographic and Psychosocial Characteristics	Comparison Group n (%)	Mood Disorder Exposure 3 and/or 6 and 18 Months n (%)	Psychotic Disorder Exposure 3 and/or 6 and 18 Months n (%)	Comorbid Anxiety and Mood Disorder Exposure 3 and/or 6 and 18 Months n (%)
**Number of mothers**	31	30	13	9
**Maternal age (years)** Median/lower quartile/upper quartile	29/25/34	30.5/27/35	34/30/35	28/26/38
**Educational achievement**				
Tertiary education ^#^	3 (9.7%)	3 (10%)	1 (7.7%)	1 (11.1%)
Secondary education ^#^	28 (90.3%)	27 (90%)	12 (92.3%)	8 (88.9%)
**Antenatal physical illness**				
HIV−	25 (80.6%)	27 (90%)	12 (92.3%)	7 (77.8%)
HIV+	6 (19.4%)	3 (10%)	1 (7.7%)	2 (22.2%)
**Antenatal substance use**				
None	25 (83.3%)	18 (60%)	7 (53.8%)	7 (77.8%)
Cigarette/tobacco smoking	0 (0%)	8 (26.7%)	4 (30.8%)	1 (11.1%)
Alcohol/recreational drugs	5 (16.7%)	4 (13.3%)	2 (15.4%)	1 (11.1%)
**Substance use (18 months)**				
None	23 (74.2%)	15 (50%)	8 (61.5%)	5 (55.6%)
Tobacco smoking	3 (9.7%)	10 (33.3%)	3 (23.1%)	2 (22.2%)
Alcohol/recreational drugs	5 (16.1%)	5 (16.7%)	2 (15.4%)	2 (22.2%)
**Psychotropic medication** **use**				
18 months	0 (0%)	22 (73.3%)	12 (92.3%)	4 (44.4%)
**Marital status (18 months)**				
Married/cohabiting	18 (58.1%)	14 (46.7%)	7 (53.8%)	4 (44.4%)
Separated/divorced/single	13 (41.9%)	16 (53.3%)	6 (46.2%)	5 (55.6%)
**Employed (18 months)**				
Employed	14 (45.2%)	12 (40%)	2 (15.4%)	4 (44.4%)
Unemployed	17 (54.8%)	18 (60%)	11 (84.6%)	5 (55.6%)
**Average family income (per month at 18 months)**				
>300 USD	5 (16.1%)	3 (10%)	0 (0%)	0 (0%)
<300 USD	26 (83.9%)	27 (90%)	13 (100%)	9 (100%)

HIV, human immunodeficiency virus; USD, United States dollars; ^#^ Completed secondary or tertiary education.

**Table 2 ijerph-20-06192-t002:** Toddler socio-demographic and health characteristics (n = 83).

Toddler Demographic and Psychosocial Characteristics	Comparison Group n (%)	Mood Disorder Exposure 3 and/or 6 and 18 Months n (%)	Psychotic Disorder Exposure 3 and/or 6 and 18 Months n (%)	Comorbid Anxiety and Mood Disorder Exposure 3 and/or 6 and 18 Months n (%)
**Number of Toddlers**	31	30	13	9
**Birth outcomes**				
**Gender**: Male	16 (51.6%)	17 (56.7%)	8 (61.5%)	4 (44.4%)
**Gender**: Female	15 (48.4%)	13 (43.3%)	5 (38.5%)	5 (55.6%)
**Gestational age (weeks):** Median/lower quartile/upper quartile	40/38/40	39/38/39	39/38/40	38/38/39
**Preterm birth (<37 weeks)**	0 (0%)	4 (13.3%)	1 (7.7%)	1 (11.1%)
**Birth weight, grams:** Median/lower quartile/upper quartile	3185/3040/3705	3027.5/2760/3320	2900/2820/3100	2785/2620/3075
**APGAR score at 5 min** Median/lower quartile/upper quartile	10/9/10	10/9/10	10/9/10	10/9/10
**Toddler 18-month outcomes**				
Age at 18-month assessment (months) Median/lower quartile/upper quartile	18.7/18.1/19.6	18.8/18.1/20.1	18.5/18.0/19.6	18.7/17.9/19.9
WAZ Median/lower quartile/upper quartile	0.86/0.47/1.19	0.39/-0.545/1.55	0.57/0.07/1.17	1.42/0.1/1.69
Low WAZ (−2 or below)	0 (0%)	0 (0%)	0 (0%)	0 (0%)
Head circumference (cm) Median/lower quartile/upper quartile	49/48/50	48.5/47/49	48/47/48	48/46.5/48
HCAZ Median/lower quartile/upper quartile	1.14/0.34/2.16	0.59/0.18/2.01	1.1/0.37/1.24	0.48/0.44/1.2
Low HCAZ (−2 or below)	0 (0%)	0 (0%)	0 (0%)	0 (0%)
Baby illness 6–18 months	0 (0%)	0 (0%)	1 (7.7%)	0 (0%)
**Current feeding practices (18 months)**				
Mixed feeding (formula and breastfeeding) and solids	29 (93.5%)	28 (93.3%)	11 (84.6%)	5 (55.6%)
Solids only	2 (6.5%)	2 (6.7%)	2 (15.4%)	4 (44.4%)
**Caregiver during the day (18 months)**				
Mother only	30 (96.8%)	29 (96.7%)	11 (84.6%)	9 (100%)
Mother and other (grandmother, nanny, crèche)	1 (3.2%)	1 (3.3%)	2 (15.4%)	0 (0%)
**Number of older siblings**				
None	9 (30%)	6 (20%)	4 (30.8%)	4 (50%)
≤2	17 (56.7%)	19 (63.3%)	7 (53.8%)	4 (50%)
≥3	4 (13.3%)	5 (16.7%)	2 (15.4%)	0 (0%)

WAZ, weight-for-age z score; HCAZ, head circumference-for-age z score.

**Table 3 ijerph-20-06192-t003:** Comparison of exposure to different groups of persistent maternal mental health disorders according to the BSID-III domain subtest composite and scaled scores at 18 months.

BSID-III Subtest and Scaled Scores at 18 Months	Comparison Group n = 31			Mood Disorder Exposure 3 and/or 6 and 18 Months n = 30				Psychotic Disorder Exposure 3 and/or 6 and 18 Months n = 13				Comorbid Anxiety and Mood Disorder Exposure 3 and/or 6 and 18 Months n = 9			
	Median	IQR	Min-Max	Median	IQR	Min-Max	*p*-Value *	Median	IQR	Min-Max	*p*-Value **	Median	IQR	Min-Max	*p*-Value ***
**Cognitive composite score**	105	95–110	85–125	105	100–110	80–125	0.861	100	95–115	75–125	0.491	110	110–115	100–120	0.049
**Motor composite score**	100	94–103	85–115	100	97–103	61–124	0.769	97	94–103	67–110	0.252	107	103–110	100–112	0.013
**Fine motor scaled score**	11	10–12	8–15	11	10–12	3–15	0.296	10	10–11	6–13	0.134	12	12–12	11–14	0.041
**Gross motor scaled score**	9	8–10	6–12	9	8–10	3–12	0.325	9	8–9	3–11	0.749	9	9–10	9–11	0.043
**Language composite score**	100	97–106	89–118	100	94–106	74–118	0.845	97	94–103	77–109	0.207	109	103–109	97–121	0.041
**Receptive scaled score**	10	9–11	8–14	11	10–11	6–14	0.341	10	10–11	5–12	0.749	11	10–12	8–15	0.091
**Expressive scaled score**	10	9–11	8–13	10	9–11	5–12	0.353	9	8–11	7–12	0.093	11	10–12	9–12	0.094
**Social-emotional composite score**	120	110–120	95–135	115	110–120	90–135	0.301	120	110–120	95–130	0.906	120	115–125	110–130	0.277
**Adaptive behaviour composite score**	93	87–105	76–114	102.5	96–106	55–116	0.131	93	95–101	63–105	0.236	100	98–106	96–114	0.058

BSID-III, Bayley Scales of Infant and Toddler Development^®^, Third Edition; IQR, interquartile range. Between-group analysis (Kruskal–Wallis test): *p*-value * for mood disorder exposure versus the comparison group; *p*-value ** for psychotic disorder exposure versus the comparison group; *p*-value *** for anxiety disorder exposure versus the comparison group.

**Table 4 ijerph-20-06192-t004:** The number of toddlers (n = 15) demonstrating a clinically delayed and/or severely impaired performance on the BSID-III domains and subscales.

	Toddler Number	Cognitive Composite Score	Motor Composite Score	Fine Motor Scaled Score	Gross Motor Scaled Score	Language Composite Score	Receptive Scaled Score	Expressive Scaled Score	Social-Emotional Composite Score	Adaptive Behaviour Composite Score
**Comparison group** **(n = 4)**	**1**	≥85	≥85	≥8	≤7	≥85	≥8	≥8	≥85	70–84
	**2**	≥85	≥85	≥8	≤7	≥85	≥8	≥8	≥85	70–84
	**3**	≥85	≥85	≥8	≤7	≥85	≥8	≥8	≥85	≥85
	**4**	≥85	≥85	≥8	≤7	≥85	≥8	≥8	≥85	≥85
**Mood disorder exposure** **(n = 8)**	**1**	≥85	70–84	≤7	≤7	≥85	≥8	≥8	≥85	70–84
	**2**	70–84	≤69	≤7	≤7	70–84	0–7	≤7	≥85	≤69
	**3**	≥85	≥85	≥8	≥8	≥85	≥8	≤7	≥85	70–84
	**4**	≥85	≥85	≥8	≥8	≥85	≥8	≤7	≥85	≥85
	**5**	≥85	≥85	≥8	≥8	≥85	≥8	≤7	≥85	≥85
	**6**	≥85	≥85	≥8	≤7	≥85	≥8	≥8	≥85	≥85
	**7**	≥85	≥85	≥8	≥8	≥85	≥8	≤7	≥85	≥85
	**8**	≥85	≥85	≥8	≥8	≥85	≥8	≤7	≥85	≥85
**Psychotic disorder exposure** **(n = 3)**	**1**	≥85	≥85	≥8	≤7	70–84	≤7	≤7	≥85	≤69
	**2**	≥85	≥85	≥8	≥8	≥85	≥8	≤7	≥85	70–84
	**3**	70–84	≤ 69	≤7	≤7	≥85	≥8	≥8	≥85	≤69
**Comorbid anxiety and mood disorder (n = 0)**	**0**	≥85	≥85	≥8	≥8	≥85	≥8	≥8	≥85	≥85

Note: ≥85 and ≥8: average or above-average performance; 70–84: delayed performance; ≤69: severely impaired performance; ≤7: below-average scaled scores. Toddlers scoring ≤ 1 SD below the mean of the composite (<85) or scaled (≤7) scores were classified as demonstrating a clinically significant developmental delay.

## Data Availability

The data are not publicly available due to ethical and healthcare privacy concerns.

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
