# Peer review of "Persistent Maternal Mental Health Disorders and Toddler Neurodevelopment at 18 Months: Longitudinal Follow-up of a Low-Income South African Cohort"

_ijerph, 2023, doi:10.3390/ijerph20126192_

Round 1

Reviewer 1 Report

It was a delight to read such a well-presented and original article assessing the relationship between maternal mental health disorders and toddler neurodevelopment in a South African cohort. The manuscript addresses a clear gap in the literature that examines such relationships in low income countries and reveals some interesting preliminary results around maternal clinical outcomes and toddler neurodevelopment. Please see specific comments and suggestions for edits to the manuscript prior to publication below. 

1.       Lines 71-82: Within this paragraph, there are a number of references to persistent mental health or psychotic disorders. Could you define ‘persistent’? How long does a disorder need to last or at what severity, until it is deemed persistent?

2.       Lines 83-84: Similarly to the previous comment, could you define maternal mood disorders and maternal psychotic disorders? Which disorders do each of these encompass? Where is the cut-off between mood and psychotic disorder?

3.       Line 113: it is assumed the justification for these exclusion criteria is that these conditions/circumstances affect child neurodevelopment (and so are confounding). Could you make the justification explicit?

4.       Line 155: could you state what these toddler risk factors were?

5.       Lines 188-189: Similarly to review comment (2), could you define maternal mood disorders and maternal psychotic disorders? Which disorders do each of these encompass? Where is the cut-off/what is the diagnostic difference between a mood and psychotic disorder?

6.       Table 1 – very informative data. It may also be insightful to see the breakdown of breast vs. formula feeding within each group if you should have these data?

7.       Table 1 – For your caregiver during the day data, are the ‘other’ caregivers definitively the ones listed – grandmothers, nanny and creche? If so, this suggests there is no father, grandfather or male daytime care involvement in your sample? This is perhaps a point worth touching on in your discussion.

8.       Page 14, line 99 – could you remove the double negative to allow for clearer understanding of this point?

9.       Lines 134-136 of Discussion: Perhaps you could consider that maternal anxiety may have adaptive benefits (i.e., increased awareness, hypervigilance, heightened nervous system, worry around e.g. safety) that may actually benefit infants.

10.   The study sample consists of mothers and infants, however I would urge you to include some reference to or discussion around the role of fathers somewhere in the manuscript. The role of the father is not captured in the study (understandably so as it was not the study aim or sample population), but infant outcomes are very much correlated with the quality of the father-child relationship too. Perhaps this could be a suggestion for future research or a potential confound or protective factor in the current study. It is important to acknowledge the presence and potential impact of the father within parent-child research to help raise the profile of positive father involvement and promote the cultural shift towards a parent representing mother and/or father, not just mother, as outdated traditions dictate.

Author Response

Thank you for your time and very valuable and constructive comments. We sincerely appreciate your comments and suggestions for revising the paper and have provided additional information (please see attached file) and in the annotated manuscript to address your concerns.

Reviewer 2 Report

Dear Authors,

Thank you for letting me review this paper. It is a very interesting topic. I do have some concerns.

-          I miss a definition for a mental health disorder.

-          What happened with pregnant women younger than 18 years old?

-          Please discuss the long-time span (2014-2019) for collecting participants.

-          How did COVID-19 affect mothers with toddlers bord in 2019?

-          How was the trained research staff trained?

-          How did you translate all the instruments to the different languages?

-          Why was not the frequency of substance not recorded?

-          Is it important to know about the mother’s HIV status?

-          Why did not the other parents been investigated?

Author Response

Thank you for your time and very valuable and constructive comments. We sincerely appreciate your comments and suggestions for revising the paper and have provided additional information (in the attached file) and in the annotated manuscript to address your concerns.

Reviewer 3 Report

Dear authors,

The research carried out reveals the need to expand the study of protective and risk factors in the perinatal period, especially in vulnerable populations. The role of mothering in the extended family as a protective factor is very interesting and we observe it very often in the clinical practice.

In relation to the exposition of the article, I would like to suggest:

1. Introduction: I would move lines 71-82 to the method section, integrating them into 2.1 Design and setting.

2. Materials and methods:

2.1 Design and setting: lines 93 (from pregnant women…) to 101 could be part of the description of the participants. This section could be also condensed and joined to 2.6 of the description

2.2 Measures: it could be reorganized contemplating within this section:

2.2.1 The interview with the socio-demographic variables

2.2.2 Bayley

2.2.3 Assessment

3. Results: I would suggest dividing table 1 into two different tables to facilitate the comprehension of the results to the reader. Perhaps it could be Table 1 Description of mothers (and families) and in a second table Infant information.

Please review table 2 comas in the p value on psychothic disorder. 

As suggested in table 3, in the group of mood disorder exposure, appeared to be more children with some domain affected. Could this be included? As mentioned in Glover research the impact of stress and mood or anxiety is different in every child. But it is half of the affected child “in a clinical delay or severely impaired” are in this mood exposure group. There are no differences in means but could it be studied the number of kids with some affected domain and compare this number with comparison group?

4. Discussion: A wide variety of studies and literature have described the antenatal impact of anxiety, depression, and stress on the neurocognitive development of fetuses. Vivette Glover affirms that this can be mitigated by a good mothering in the postpartum period. Given that it is a population with very specific characteristics, the hypothesis that multifamily structures may contribute to this factor described by Glover seems very suggestive to me. I think this aspect of the wide mothering function developed in the extended family as a protective factor could be highlighted and perhaps included in future research. Thus, it could be pointed out in the abstract and in the conclusions as hypotheses that allow us to understand the results.

Author Response

(The authors gave the same response as above.)
